# Fibrin Network Formation and Lysis in Septic Shock Patients

**DOI:** 10.3390/ijms22179540

**Published:** 2021-09-02

**Authors:** Julie Brogaard Larsen, Mathies Appel Aggerbeck, Kim Michael Larsen, Christine Lodberg Hvas, Anne-Mette Hvas

**Affiliations:** 1Thrombosis and Haemostasis Research Unit, Department of Clinical Biochemistry, Aarhus University Hospital, Palle Juul-Jensens Boulevard 99, DK-8200 Aarhus, Denmark; mathies_appel@live.dk (M.A.A.); am.hvas@dadlnet.dk (A.-M.H.); 2Department of Anaesthesiology and Intensive Care, Aarhus University Hospital, Palle Juul-Jensens Boulevard 99, DK-8200 Aarhus, Denmark; kim.m.larsen@midt.rm.dk (K.M.L.); c.hvas@clin.au.dk (C.L.H.); 3Department of Clinical Medicine, Aarhus University, Palle Juul-Jensens Boulevard 82, DK-8200 Aarhus, Denmark

**Keywords:** sepsis, fibrinolysis, fibrin clot lysis time, plasminogen, plasminogen activator inhibitor 1, disseminated intravascular coagulation

## Abstract

Background: Septic shock patients are prone to altered fibrinolysis, which contributes to microthrombus formation, organ failure and mortality. However, characterisation of the individual patient’s fibrinolytic capacity remains a challenge due to a lack of global fibrinolysis biomarkers. We aimed to assess fibrinolysis in septic shock patients using a plasma-based fibrin clot formation and lysis (clot–lysis) assay and investigate the association between clot–lysis parameters and other haemostatic markers, organ dysfunction and mortality. Methods: This was a prospective cohort study including adult septic shock patients (*n* = 34). Clot–lysis was assessed using our plasma-based in-house assay. Platelet count, activated partial thromboplastin time (aPTT), international normalised ratio (INR), fibrinogen, fibrin D-dimer, antithrombin, thrombin generation, circulating fibrinolysis markers and organ dysfunction markers were analysed. Disseminated intravascular coagulation score, Sequential Organ Failure Assessment (SOFA) score and 30-day mortality were registered. Results: Three distinct clot–lysis profiles emerged in the patients: (1) severely decreased fibrin formation (flat clot–lysis curve), (2) normal fibrin formation and lysis and (3) pronounced lysis resistance. Patients with abnormal curves had lower platelet counts (*p* = 0.05), more prolonged aPTT (*p* = 0.04), higher lactate (*p* < 0.01) and a tendency towards higher SOFA scores (*p* = 0.09) than patients with normal clot–lysis curves. Fibrinogen and fibrin D-dimer were not associated with clot–lysis profile (*p* ≥ 0.37). Conclusion: Septic shock patients showed distinct and abnormal clot–lysis profiles that were associated with markers of coagulation and organ dysfunction. Our results provide important new insights into sepsis-related fibrinolysis disturbances and support the importance of assessing fibrinolytic capacity in septic shock.

## 1. Introduction

Septic shock patients are at high risk of developing coagulation disturbances, which, in their most severe form, may manifest as disseminated intravascular coagulation (DIC), a life-threatening condition [1,2]. Altered fibrinolysis is recognised as a contributor to sepsis-related coagulopathy and may aggravate microthrombus formation, organ failure and mortality [3,4,5]. Sepsis patients are generally considered to have impaired fibrinolysis [3,4]; however, recent literature indicates that some patients with sepsis-related DIC have decreased fibrin formation capacity [6] and even enhanced clot susceptibility to lysis [7], and hypocoagulability in sepsis has been associated with poor prognosis [8]. This may hold new implications for the treatment of coagulation disturbances in sepsis, as patients with decreased fibrin formation capacity or increased lysis could benefit from antifibrinolytics in the case of active bleeding or the need for invasive procedures. However, the laboratory tests currently used for coagulation assessment in daily clinical practice cannot assess altered fibrin formation or fibrinolysis with sufficient sensitivity or specificity. Both fibrinogen and fibrin D-dimer are acute-phase reactants and will increase in sepsis [9], and fibrin D-dimer is influenced both by the amount of fibrin formation and the rate of fibrin breakdown and is therefore an unspecific marker for altered fibrinolysis at best. Thus, global and dynamic assays of fibrin formation and fibrinolysis are needed to investigate fibrinolysis in detail.

Several methods have been developed to assess fibrin formation and lysis (reviewed by Ilich et al. [10]). The fibrin clot formation and lysis (clot–lysis) assay is a plasma-based assay that can detect both hyper- and hypofibrinolysis [11]. Coagulation and fibrinolysis are initiated simultaneously ex vivo using tissue factor (TF) and tissue plasminogen activator (tPA), and absorbance is continuously registered. This allows the calculation of lag phase (indicating time to clot initiation), peak absorbance (indicating maximum fibrin concentration), area under curve (indicating net fibrin formation) and time to 50% lysis of the clot, denominated clot–lysis, indicating the rate of fibrinolysis and thus fibrinolytic capacity. The clot–lysis assay is employed in a few research laboratories worldwide and is mainly used as a research tool [12]. However, this assay provides detailed information about coagulation and fibrinolysis; thus, its potential in patients with potentially disturbed fibrinolysis should be explored further.

The present study aimed to investigate fibrin formation and lysis in septic shock patients and its association with other markers of coagulation and fibrinolysis as well as DIC, organ dysfunction severity and 30-day mortality, using our in-house clot–lysis assay [11].

## 2. Results

### 2.1. Study Population

The present study was a post-hoc study to a prospective cohort study including adult septic shock patients admitted to the intensive care unit (ICU) [13]. Citrated plasma for the clot–lysis assay was available for 34 of the 36 patients (94%) enrolled in the original study population on day 1. Demographic and clinical characteristics of the 34 patients are displayed in Table 1. In general, the patients were elderly, with a mean age of 68 (standard deviation (SD) 15) years, and the majority (74%) were male. The patients displayed a pronounced acute-phase response with high plasma levels of C-reactive protein, fibrinogen and fibrin D-dimer. Sequential Organ Failure Assessment (SOFA) scores were moderately high, with a median of 11. Eleven patients (32%) had DIC score ≥ 5. Mortality at 30 days was approximately 20%.

### 2.2. Clot–Lysis Assay

Three distinct clot–lysis profiles emerged in our patient group: (1) severely decreased fibrin formation (flat curve), (2) normal fibrin formation and lysis (normal curve) and (3) pronounced lysis resistance (lysis-resistant curve); see Figure 1A.

On day 1, 10 of the 34 patients (30%) demonstrated flat curves, with no or little fibrin formation and area under curve = 0. Another 13 (38%) patients showed normal curves, with normal or increased fibrin formation and full lysis of the clot. Nine of these 13 patients had an increased area under curve when using our local reference intervals [14] as cut-off, indicating increased net fibrin formation, but only two had prolonged clot–lysis times compared with reference intervals (Table 2). The remaining 11 (32%) patients had lysis-resistant curves, showing normal or increased fibrin formation but pronounced resistance to lysis, with no or only partial clot–lysis after 90 min of reading.

On days 2 and 3, 29 and 23 patients remained included in the cohort, with plasma available for analysis. On both days, the three distinct curve patterns remained (Table 3). On day 2, 15 patients (52%) had flat curves, 7 patients (24%) had normal curves, and 7 patients (24%) had lysis-resistant curves, while on day 3, 8 patients (35%) had flat curves, 10 patients (43%) had normal curves, and 5 patients (22%) had lysis-resistant curves. When investigating the consistency of the clot–lysis pattern at an individual patient level, we found that of the 29 patients remaining in the cohort on day 2, 22 (76%) exhibited the same clot–lysis pattern on day 2 as on day 1, and of the 23 patients remaining on day 3, 14 (61%) exhibited the same clot–lysis pattern on day 3 as on day 1.

### 2.3. Clot–Lysis Profile and Coagulation Parameters

In general, patients with abnormal clot–lysis curves (flat or lysis-resistant curves) had lower platelet counts and more prolonged activated partial thromboplastin time (aPTT) (Figure 1B) than patients with normal curves. Median with interquartile range (IQR) for platelets were: flat curves 106 (68–199) × 10^9^/L, normal curves 247 (175–299) × 10^9^/L, lysis-resistant curves 152 (111–245) × 10^9^/L, *p* = 0.05 (Kruskal–Wallis); see Figure 1B. Subsequent statistic tests of differences between individual groups showed significant differences in platelet counts between patients with flat and normal curves (*p* = 0.02), but not between patients with normal curves and lysis-resistant curves (*p* = 0.17). Median (IQR) for aPTT between the three patient groups were: flat curves 41 (37–50) seconds, normal curves 32 (27–40) seconds, lysis-resistant curves 33 (27–39) seconds, *p* = 0.04 (Kruskal–Wallis). Fibrinogen, fibrin D-dimer, international normalised ratio (INR) and antithrombin did not differ between patients with different clot–lysis curves (Figure 1B). Fibrin D-dimer correlated poorly with 50% clot–lysis time in the 13 patients with normal curves (Spearman’s rho −0.08, *p* = 0.79).

Ex vivo thrombin generation was analysed using Calibrated Automated Thrombogram^®^ (CAT). As previously described [13], approximately 30% of the patients (*n* = 10) displayed flat thrombin generation curves with endogenous thrombin potential (ETP) being 0, indicating severely impaired thrombin generation. There was a considerable overlap between patients with flat thrombin generation curves and those with flat clot–lysis curves, as 8 out of 10 patients with flat clot–lysis curves also had flat thrombin generation curves (*p* < 0.001, Chi-square test). Peak thrombin concentration tended to be higher in patients with lysis-resistant curves than patients with normal clot–lysis curves (mean 216 (SD 77) nmol/L vs. 151 (SD 77) nmol/L, *p* = 0.06), but ETP did not differ between patients with normal clot–lysis curves and those with lysis-resistant curves (data not shown).

Peak fibrin formation (peak absorbance) correlated strongly with plasma fibrinogen (Spearman’s rho = 0.77, *p* > 0.001) and moderately with peak thrombin concentration (Spearman’s rho = 0.59, *p* > 0.01). Net fibrin formation (area under curve) correlated moderately with plasma fibrinogen (Pearson’s r = 0.56, *p* = 0.05) but not with peak thrombin concentration (Pearson’s r = 0.19, *p* = 0.53) or with ETP (Pearson’s r = 0.10, *p* = 0.70).

### 2.4. Clot–Lysis Profile and Circulating Fibrinolysis Markers

As displayed in Figure 2, plasma levels of plasminogen were lower and levels of plasminogen activator inhibitor (PAI)-1 were higher in lysis-resistant patients than in patients with normal clot–lysis curves on day 1 (plasminogen: median (IQR) 63 (50–72)% vs. 78 (62–97)%, *p* = 0.02; PAI-1: median (IQR) 466 (202–700) µg/L vs. 79 (57–173) µg/L, *p* < 0.01). Similar results were found on day 2, where median PAI-1 was 168 (IQR: 57–365) µg/L in lysis-resistant patients (*n* = 6) vs. 38 (IQR: 19–49) µg/L in patients with normal curves (*n* = 7). Levels of thrombin-activatable fibrinolysis inhibitor (TAFI) and tissue plasminogen activator (tPA) did not differ between patients with normal and lysis-resistant curves on day 1. In patients with normal curves (*n* = 13), where 50% lysis time could be calculated, PAI-1 correlated positively and significantly with 50% clot–lysis time on day 1 (Spearman’s rho = 0.72, *p* < 0.01). We found only weak and non-significant correlations between 50% clot–lysis time and plasminogen, tPA, TAFI or fibrin D-dimer on day 1 (rho values −0.08 to 0.36, *p* all > 0.20).

### 2.5. Clot–Lysis Profile and Organ Dysfunction

We observed higher SOFA scores and arterial lactate in patients with abnormal clot–lysis curves than in patients with normal curves (Figure 3). Median (IQR) SOFA scores in the three patient groups were: flat curves 13 (10–15) points, normal curves 8 (7–11) points, lysis-resistant curves 11 (8–15) points, *p* = 0.09 (Kruskal–Wallis test). The difference between patients with flat and normal curves reached statistical significance (*p* = 0.03 (Mann–Whitney test)). Median (IQR) lactate in the three patient groups were: flat curves 2.5 (1.6–6.0) mmol/L, normal curves 1.2 (1.0–1.4) mmol/L, lysis-resistant curves 2.8 (1.4–4.7) mmol/L, *p* < 0.01 (Kruskal–Wallis test). Patients with flat or lysis-resistant curves appeared to have higher DIC scores than patients with normal curves, but this was not statistically significant (Figure 3).

## 3. Discussion

We investigated fibrinolysis in septic shock patients using our in-house fibrin clot formation and lysis assay as well as circulating markers of fibrinolysis. The main finding was the presence of three distinct clot–lysis patterns during sepsis: a group of patients with flat curves indicating no fibrin formation, a group with lysis-resistant curves despite high tPA concentrations in the assay and, finally, a patient group with increased net fibrin formation but generally normal clot–lysis capacity.

Lysis resistance appeared to be driven mainly by increased PAI-1, with low plasminogen as a possible contributor. In the present study, TAFI was not associated with clot–lysis pattern, contrary to the findings of Semeraro et al. [15]. However, the TAFI assay used in the present study measures total TAFI and not activated TAFI; this may explain the discrepancy, as total TAFI may decrease during sepsis due to consumption [15]. The remaining patients had normal or increased net fibrin formation but, notably, were able to lyse their clots completely, and almost all had normal 50% clot–lysis time compared with our age-specific reference interval [14]. Fibrin D-dimer generally showed a pronounced elevation regardless of clot–lysis curve pattern and did not differ between patients with lysis resistance and patients with flat or normal curves, nor did it correlate with 50% clot–lysis time in the 13 patients where lysis time could be calculated. This is as expected, since circulating concentrations of fibrin D-dimer are dependent on both the amount of intravascular fibrin formation and the rate of fibrin breakdown. Our results clearly demonstrate that fibrin D-dimer is not a reliable marker for fibrinolytic capacity.

The flat clot–lysis curves found in approximately one third of the patients were probably explained by reduced thrombin activity, as these patients had low ex vivo thrombin generation and prolonged aPTT compared with other patients. Of note, neither INR nor fibrinogen revealed a decreased fibrin formation capacity. INR was similar between patients with flat curves and patients with normal curves, and fibrinogen was within or above reference intervals for all but one patient. Patients with flat clot–lysis curves had lower platelet counts, higher lactate and higher SOFA scores than patients with normal curves, indicating more severe sepsis and organ dysfunction. We excluded patients receiving anticoagulant treatment in the form of vitamin K antagonists or direct oral anticoagulants. However, as patients received thromboprophylaxis in the form of low-molecular-weight heparin (LMWH) in accordance with local institutional guidelines, we cannot exclude that this could influence fibrin formation capacity and contribute to the flat curves in some of the patients. However, all patients received LMWH during their ICU stay and it was dosed in the evening, while blood sampling was performed in the morning. Unfortunately, we could not measure anti-Xa activity due to lack of material.

The presence of three distinct clot–lysis patterns remained on days 2 and 3. Furthermore, we found that 75% of patients exhibited the same pattern on day 2 as on day 1, while on day 3, this number had decreased to 61%. The change from day 1 to day 3 probably reflects that sepsis-related coagulopathy and altered fibrinolysis are dynamic conditions that may fluctuate during the course of disease. However, the high concordance between day 1 and day 2 supports the validity of our results. It should be noted, however, that some patients dropped out of the study between day 1 and 2 or day 2 and 3 due to discharge from ICU, transfer to different hospitals or passing, and we do not have access to data from these patients; thus, these concordance rates may be subject to bias and should be interpreted with caution.

A detailed assessment of fibrin formation and lysis capacity in critically ill patients can support the early diagnosis of microthrombus formation, but it may also benefit the bleeding patient. Septic shock patients are at high risk of bleeding due to low platelet counts, consumption coagulopathy and higher need for invasive procedures such as lumbar puncture, surgical exploration and vascular access for, e.g., haemodialysis or central venous catheters. Currently, haemostatic support with tranexamic acid is considered contraindicated in sepsis, especially when DIC is suspected [16]. However, tranexamic acid could be highly beneficial for the patient with normal fibrinolytic capacity and high bleeding risk [17]. Furthermore, PAI-1 inhibitors are currently emerging as a potential treatment in patients with microthrombus formation due to impaired fibrinolysis [18,19]. Reliable and quick assessment of fibrinolytic capacity will be pivotal in supporting the testing and implementation of these agents in clinical practice.

Four recent studies assessed fibrin clot properties in sepsis with clot–lysis assays similar to ours [15,20,21,22]. Blasi et al. investigated patients with hepatic dysfunction and found prolonged lysis times in septic vs. non-septic patients and reported that impaired fibrinolysis was associated with organ dysfunction severity [20]. Semeraro et al. assessed fibrinolysis in septic shock patients stratified by platelet count [15]. Interestingly, they reported that prolonged clot–lysis time was associated with low platelet counts, similar to our findings. Very recently, Lisman et al. compared clot–lysis between 31 acute-on-chronic liver failure patients and 20 sepsis patients with normal hepatic function and found normal to prolonged lysis time in sepsis patients, despite similar levels of PAI-1 and plasmin–antiplasmin complex in the two groups [21]. Finally, Bouck et al. investigated fibrinolysis in new coronavirus disease (COVID-19) and non-COVID-19 sepsis patients [22]. Both COVID-19 and non-COVID-19 sepsis patients had increased fibrin formation compared with healthy individuals, and non-COVID-19 sepsis patients had prolonged lysis times compared with healthy controls. Of note, both fibrin formation and lysis times varied widely in the sepsis group. We add to current knowledge in the field by providing a detailed description of coagulation and fibrinolysis parameters and by describing associations between fibrin formation, lysis patterns and patient characteristics, including additional haemostatic parameters and organ dysfunction.

The clot–lysis assay appears well suited for characterisation of the fibrin formation and fibrinolytic capacity in septic shock patients and holds potential as a research assay, as it is sensitive to both hyper- and hypofibrinolysis, can be run in batch and can be modified according to different clinical research questions. However, turnover time, lack of standardisation or harmonisation between laboratories [12] and lack of automation currently challenge the utility of the assay in the routine laboratory setting. These questions should be addressed in order for the assay to be useful for daily clinical practice. Viscoelastic tests are attractive alternatives as they are semi-automated, have fast turnover times and include the contribution of platelets and erythrocytes to clot formation. Sepsis patients have been shown to have a significantly higher lysis index at 45 and 60 min (LI45 and LI60) than healthy individuals when measured with rotational thromboelastometry (ROTEM) [23]. However, standard viscoelastic protocols are not very sensitive to hypofibrinolysis, as these protocols typically have narrow reference intervals for lysis indices, with upper reference limits close to or including 0% lysis. An alternative could be modified viscoelastic testing with added tPA or urokinase and low TF concentrations. This type of assay has been employed in sepsis patients by Kuiper et al. [24] and Panigada et al. [7] with promising results and in other patient populations by our group [25]. The next step should be the investigation and validation of both the plasma clot–lysis assay and modified viscoelastic tests in larger intensive care cohorts.

Some limitations to the present study should be considered. It was a post-hoc study with a small sample size. The study was not powered to investigate clinical endpoints such as bleeding, thrombosis and mortality. We did not have access to arterial lactate levels at the time of ICU admission but only at study enrolment and thus were not able to take arterial lactate at admission into account. The use of different TF and tPA concentrations challenges the comparison of our results with those of others. Our clot–lysis protocol uses relatively high concentrations of tPA compared with other laboratories [15,20]. Even so, lysis resistance was observed in one third of patients, and the assay was sensitive to variation in net fibrin formation and 50% clot–lysis time in those patients who obtained full clot–lysis. However, the high tPA concentration may render the protocol less sensitive to endogenous tPA, which could limit its use in populations with a high prevalence of hyperfibrinolysis, e.g., severe trauma and obstetric DIC.

## 4. Materials and Methods

### 4.1. Design and Study Population

The present study was a post-hoc study of a prospective cohort study [13] including septic shock patients ≥ 18 years admitted to the ICU, Aarhus University Hospital, Aarhus, Denmark, between October 2016 and February 2018. Patients were enrolled within 24 h of ICU admission and a blood sample was obtained on the first morning after ICU admission (day 1), and on the morning of day 2 and day 3 if the patient was still in the ICU at these time points. Sepsis was defined according to the Sepsis-3 guidelines as clinical suspicion of infection by the admitting physician and a SOFA score ≥ 2 [26]. Shock was defined as need for noradrenaline treatment to maintain mean arterial pressure > 65 mmHg. Exclusion criteria were pregnancy, active cancer or chemotherapy within three months, major trauma or surgery within 24 h, known congenital bleeding disorder or thrombophilia, plasma transfusion within the last three days and treatment with vitamin K antagonists or direct oral anticoagulants. In accordance with local institutional guidelines for thromboprophylaxis in severe sepsis, the vast majority of patients received LMWH in the form of dalteparin, 5000 IU once daily, which was dosed in the evening, i.e., approximately 12 h prior to blood sampling. Demographic and clinical information (age, sex, comorbidities, infection focus, suspected pathogen, vital parameters including SOFA score variables [27] and 30-day mortality) was obtained from medical records. DIC score was calculated according to the International Society on Thrombosis and Haemostasis [28]. Follow-up for mortality was performed at day 30 for all patients via the electronic patient record. The project was approved by the local institutional board and the Danish Data Protection Agency. According to the Danish Health Care Act, requirement for written informed consent was waived after formal review by the regional Health Ethics Committee (file no 1-16-02-505-16). The study was performed in accordance with the Helsinki Declaration and the Danish Health Care Act.

### 4.2. Blood Sampling and Laboratory Analysis

Blood was drawn from an arterial cannula placed in the radial or femoral artery. Citrated plasma for clot formation and lysis assay, thrombin generation assay, plasma fibrinolysis markers and other coagulation analyses was obtained as previously described [13], centrifuged at 3000× *g* at 20 °C for 25 min within 1 h of sampling and stored at −80 °C until analysis within 12 months. Repeated freeze–thawing was avoided.

### 4.3. Fibrin Clot Formation and Lysis Assay

The fibrin clot formation and lysis assay (clot–lysis assay) was performed at the Thrombosis and Haemostasis Research Unit, Department of Clinical Biochemistry, Aarhus University Hospital according to our local protocol [11]. Briefly, citrated plasma was thawed at 37 °C and centrifuged for 3 min at 15,000× *g*. Final concentrations in the wells were as follows: Recombinant TF (Siemens Healthcare, Marburg, Germany), diluted 1:5000; phospholipid suspension containing phosphatidylserine, phosphatidylcholine and sphingomyelin (Rossix, Mölndal, Sweden), 4 µM; tPA (Calbiochem, San Diego, CA, USA), 116 µg/L; CaCl_2_, 26.7 mM. Normal pooled plasma (Cryocheck^TM^, Hemochrom Diagnostica, Frederiksberg, Denmark. Cat.no.: CCN-10) was used as internal control and added to all plates. After addition of CaCl_2_ as the final step, the plate was shaken for 10 s. Absorbance was read continuously at 405 nm for 90 min (Victor microplate reader, Perkin Elmer, Waltham, MA, USA). The analyses were performed in duplicate, and the mean value of the duplicate readings was used for statistics. The raw data (absorbance units (AU) with corresponding time points) were exported from the reader and processed using the 2030 WorkOut and WorkOut 2.5 software (Perkin Elmer, Waltham, MA, USA). The following parameters were derived from the raw data: lag phase (seconds), peak absorbance (AU), area under curve (AU*seconds), 50% clot–lysis time (seconds) calculated as time from peak absorbance until 50% lysis (Figure 4).

Laboratory analyses and quality assessment were performed by experienced laboratory technicians blinded to the patients’ clinical and laboratory information. Individual patient samples were re-analysed if the intra-individual (between duplicates) coefficient of variation of any parameter was >15%. The plate was re-analysed if the intra-assay (between controls) coefficient of variation of any parameter was >15%.

Our previously published reference intervals for the method were used for comparison of septic shock patients with healthy individuals [14].

### 4.4. Circulating Fibrinolysis Markers

The following proteins were analysed using commercial ELISA kits according to the manufacturers’ instructions: tPA antigen (Technozyme^®^, Technoclone, Vienna, Austria), PAI-1 antigen (Technozyme^®^, Technoclone, Vienna, Austria) and TAFI antigen (Imuclone™, Biomedica Diagnostics, Windsor, ON, Canada). Plasminogen was analysed employing a Sysmex CS2100i (Sysmex, Kobe, Japan).

### 4.5. Thrombin Generation

Ex vivo thrombin generation (Calibrated Automated Thrombogram^®^ (CAT)) was analysed at the Thrombosis and Haemostasis Research Unit, Department of Clinical Biochemistry, Aarhus University Hospital as previously described [13]. Briefly, this is a dynamic assay measuring thrombin activity in plasma, expressed as conversion of a thrombin-specific substrate, after activation with TF, phospholipids and calcium. Citrated platelet-poor plasma was used. Final concentration of TF was 1 pM. The variables lag time (minutes), time to peak thrombin concentration (minutes), velocity index (nM/minute), peak thrombin concentration (nM) and ETP (nM*min) were registered.

### 4.6. Routine Coagulation Parameters, Inflammation Markers and Organ Dysfunction Markers

The following analyses were performed at the Department of Clinical Biochemistry, Aarhus University Hospital, Denmark according to ISO15189-accredited routine protocols: blood platelet count and leukocyte count, INR, aPTT, plasma fibrinogen (functional, Clauss), fibrin D-dimer, antithrombin (functional), CRP, creatinine, alanine transaminase, bilirubin and arterial blood lactate.

### 4.7. Statistical Analysis

Statistical analyses were performed using Stata^®^ 14 (StataCorp, TX, USA). Graphs were created using GraphPad^®^ Prism 8 (GraphPad Software Inc., San Diego, CA, USA). Normal distribution was assessed visually with quantile–quantile (QQ) plots. Unpaired *t*-tests or one-way analysis of variance (ANOVA) were used to test for difference between groups if data followed a normal distribution, and Mann–Whitney’s test or Kruskal–Wallis’ test if they did not. For correlation analysis, Pearson’s r was used for normally distributed data and Spearman’s rho for non-normally distributed data. As the present study was a post-hoc study with a fixed population size, we did not perform a sample size calculation.

## 5. Conclusions

Septic shock patients showed distinct fibrin clot formation and lysis profiles detected with our assay, and these profiles correlated with markers of coagulation and organ dysfunction. Our results stress the importance of employing dynamic assays to assess fibrinolytic capacity, as these assays may provide valuable information to support and guide the treatment of sepsis-related coagulopathy.

## Figures and Tables

**Figure 1 ijms-22-09540-f001:**
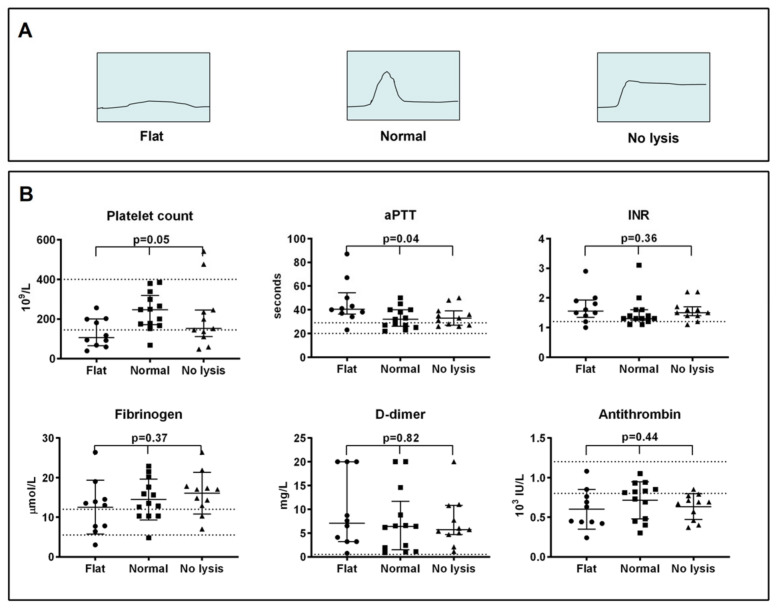
(**A**) Illustration of “flat”, “normal” and “lysis-resistant” fibrin clot formation and lysis curves; (**B**) Standard coagulation parameters in patients with flat (*n* = 10), normal (*n* = 13) and lysis-resistant (*n* = 11) curves. Median with interquartile range is shown. *p*-values indicate comparison between all three groups (one-way analysis of variance or Kruskal–Wallis test). Dotted lines indicate upper/lower reference limits as established by our laboratory. Abbreviations: aPTT, activated partial thromboplastin time; INR, international normalised ratio.

**Figure 2 ijms-22-09540-f002:**
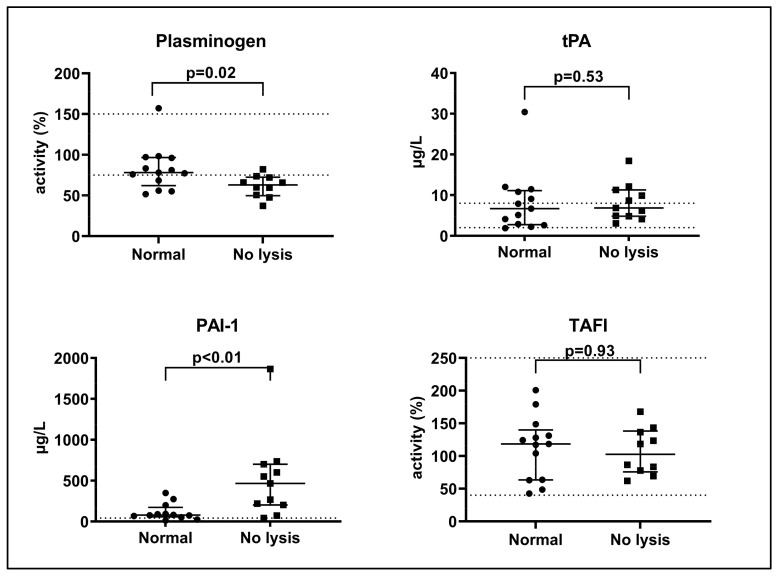
Plasminogen, tissue plasminogen activator (tPA), plasminogen activator inhibitor (PAI)-1 and thrombin-activatable fibrinolysis inhibitor (TAFI) in patients with normal lysis and lysis resistance on day 1. Median with interquartile range is shown. *p*-values indicate comparison between normal and lysis-resistant curves (*t*-test or Mann–Whitney’s test). Dotted lines indicate upper/lower reference limits as provided by the manufacturers.

**Figure 3 ijms-22-09540-f003:**
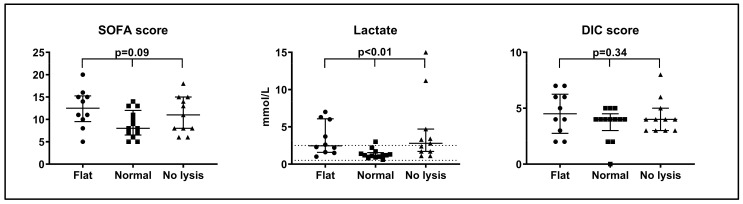
Sequential Organ Failure Assessment (SOFA) score, arterial blood lactate and disseminated intravascular coagulation (DIC) score in patients with flat, normal and lysis-resistant fibrin formation and lysis curves.

**Figure 4 ijms-22-09540-f004:**
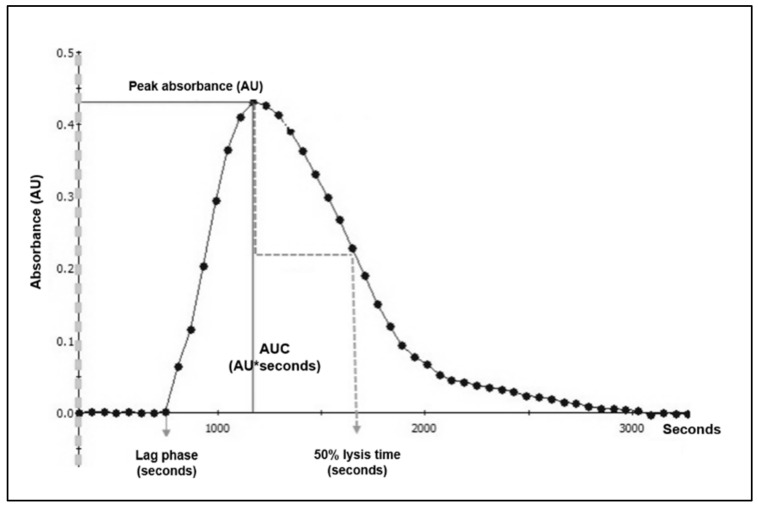
Example of normal fibrin clot formation and lysis curve in citrated plasma from a healthy individual. Abbreviations: AU, absorbance units; AUC, area under curve. (Modified from Larsen and Hvas, Methods and Protocols 20203(4):67 [11] (Open Access)).

**Table 1 ijms-22-09540-t001:** Demographic and clinical characteristics of septic shock patients (*n* = 34).

Demographic data	
Age, years; mean (SD)	67.8 ± 14.6
Sex, m/f; *n* (%)	25/9 (74%/26%)
Comorbidities; *n (%):*	
Cardiovascular disease	17 (50)
Diabetes mellitus	8 (24)
Hepatic dysfunction	4 (12)
Chronic renal insufficiency	5 (15)
**Pathogen; *n***	
Gram-positive	8
Gram-negative	7
Fungal	4
Mixed	4
Not identified	11
**Disease severity**	
30-day mortality; *n* (%)	7 (21)
ISTH DIC > 5 (day 1); *n* (%)	11 (32)
SAPS-III score; median (IQR)	64 (58–74)
SOFA score, day 1; median (IQR)	11 (8–14)
**Laboratory results, day 1 (median (IQR)**	
Lactate (arterial), mmol/L (reference interval: 0.5–2.5)	1.7 (1.2–3.3)
C-reactive protein, mg/L (reference interval: <8)	211 (109–308)
Leukocyte count, ×109/L (reference interval: 3.5–10.0)	11.6 (7.8–19.5)

Abbreviations: DIC, disseminated intravascular coagulation; f, female; IQR, interquartile range; ISTH, International Society on Thrombosis and Haemostasis; m, male; n, number of patients; SAPS, Simplified Acute Physiology Score; SD, standard deviation; SOFA, Sequential Organ Failure Assessment.

**Table 2 ijms-22-09540-t002:** Fibrin formation and lysis variables in patients with flat, normal and lysis-resistant fibrin formation curves on day 1. Medians with interquartile ranges are reported.

	Flat (*n* = 10)	Normal (*n* = 13)	Lysis-Resistant (*n* = 11)	Reference Range ^a^
Peak absorbance, AU	- ^b^	0.87 (0.82–0.91)	0.89 (0.83–1.01)	0.26–0.78
Area under curve, AU × seconds	- ^b^	1422 (1111–1666)	- ^c^	248–1130
50% lysis time, seconds	- ^b^	1042 (867–1155)	- ^c^	314–1438
Patients above reference, *n* (%):				
-Peak absorbance -Area under curve -50% lysis time	- ^b^ - ^b^ - ^b^	10 (77%) 9 (69%) 2 (15%)	9 (82%) - ^c^ - ^c^	

^a^ Established in healthy individuals > 40 years old, *n* = 66 (both sexes combined), published in [14]. ^b^ Could not be calculated from raw data, as absorbance did not increase from baseline (no fibrin formation). ^c^ Could not be calculated from raw data, as no or only partial lysis was obtained. AU, absorbance units.

**Table 3 ijms-22-09540-t003:** Patients with flat, normal or lysis-resistant fibrin formation curves on days 1, 2 and 3.

		Day 1 (*n* = 34)	Day 2 (*n* = 29)	Day 3 (*n* = 23)
Clot–lysis pattern, *n* (%)				
-Flat		10 (30%)	15 (52%)	8 (35%)
-Normal		13 (38%)	7 (24%)	10 (43%)
Lysis-resistant		11 (32%)	7 (24%)	5 (22%)
**Clot–lysis parameters**				
	**Reference**	**Sepsis patients**		
		**Day 1**	**Day 2**	**Day 3**
Peak absorbance, AU ^a^	0.26–0.78	0.89 (0.82–0.98)	0.90 (0.51–0.96)	0.72 (0.55–0.91)
Area under curve, AU × seconds ^a^	248–1130	1422 (1112–1666)	1510 (1045–1923)	1044 (787–1562)
50% lysis time, seconds ^b^	314–1438	1093 (565)	885 (313)	792 (246)

Abbreviations: AU, absorbance units. ^a^ Median with interquartile range. *n* = 24 on day 1, *n* = 14 on day 2, *n* = 15 on day 3 (patients with flat curves excluded from analysis). ^b^ Mean with SD. *n* = 13 on day 1, *n* = 7 on day 2, *n* = 10 on day 3 (patients with flat or lysis-resistant curves excluded from analysis).

## Data Availability

The data presented in this study are available on request from the corresponding author. The data are not publicly available due to privacy concerns.

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
