# Peer review of "Fibrin Network Formation and Lysis in Septic Shock Patients"

_ijms, 2021, doi:10.3390/ijms22179540_

Round 1

Reviewer 1 Report

 Fibrin network formation and lysis in septic shock patients

This original article written by Larsen and al. aims to assess fibrin formation and lysis in septic shock patients using an in-house clot-lysis assay (published protocol). Authors also assess the association of the clot-lysis assay results with other routine and research markers of coagulation and fibrinolysis, organ dysfunction severity and 30-day mortality. It is a post-hoc study of a prospective cohort study including 36 adults in septic shock admitted to the intensive care unit (Larsen JB, Laursen MA, Hvas CL, Larsen KM, Thiel S, Hvas AM. Reduced Mannose-Binding Lectin-Associated Serine Protease 416 (MASP)-1 is Associated with Disturbed Coagulation in Septic Shock. Thromb Haemost. 2019;119(6):952-61).

The paper is well written and well organized. The introduction presents the topic and its challenges, ie difficulties to identify fibrinolytic status in septic patients, which is a crucial point in order to improve sepsis management with individualized patient care. Introduction also presents the clot-lysis assay and exposes the main objectives of the research.

Results are organized in five sections including study population, results of the clot-lysis assay, clot lysis profile and coagulation parameters, clot lysis profile and fibrinolysis markers, and finally clot-lysis profile and organ dysfunction. The conclusion is in accordance with the results exposed in the paper.

Here are my main major and minor comments.

Major comments.

1/ This study provides a detailed and valuable description of various biological markers of coagulation and fibrinolytic functions and the results of an in-house clot-lysis assay at day 1 of ICU admission in septic patients. This well-conducted description is of interest for physicians and reflects the different coagulation profiles that can be encountered during sepsis, which are often unpredictable when only “routine” biological markers are available. However, I don’t think the paper provides any new information as these profiles are already known and described.

I do not agree with the authors when they say that normal lysis profile constitutes the novelty of this paper (page 7 lines 193-19.

1A/ Indeed (and fortunately), not all septic patients experiment sepsis-related coaguloapathy/DIC/altered fibrinolysis. These pathologic conditions are a continuum ranging from normal coagulation and fibrinolytic status to DIC, mainly determined by the global severity of the illness.

This is suggested by the differences between the flat curve group (higher SOFA score, elevated lactate level, DIC>=5 = 5/10: advanced sepsis? Higher saps3? Higher vasopressive amine?) compared to normal lysis curve group (normal lactate level, low SOFA score, DIC>=5 =3/13: Low severity sepsis with rapid recovery after treatment? Early sepsis? cured sepsis?). Indeed, patients fulfilling septic shock definition actually present a large disparity in the severity of the illness.

1B/Moreover, sepsis and subsequent SIC/DIC/impaired fibrinolysis are highly dynamic entities over the time and coagulation/fibrinolytic functions exhibit large variations over time, from the very beginning of the infection to sepsis /septic shock condition and finally recovery or uncontrolled disease leading to death.

For instance, in the original cohort (Larsen JB, Laursen MA, Hvas CL, Larsen KM, Thiel S, Hvas AM. Reduced Mannose-Binding Lectin-Associated Serine Protease 416 (MASP)-1 is Associated with Disturbed Coagulation in Septic Shock. Thromb Haemost. 2019;119(6):952-61), 10 patients evolved in 48hours from abnormal clot-lysis curve to normal curve. So, one of the main limit of this study is that this is not a dynamic description but only a single timepoint picture of coagulation/ fibrinolysis function in septic patients with variable severity and who might be at different timepoints of the disease at ICU admission. It could have been really interesting to analyse patients coagulation profile at day2 and 3, similar to what was done in the original paper.

Other works highlight these disparities in septic patients, leading to the different coagulation/fibrinolysis profiles, and their evolution over time (Schmitt et al. Ann. Intensive Care (2019) doi 10.1186/s13613-019-0499-6 ; Semeraro et al. CCM 2017, DOI: 10.1097/CCM.0000000000002919

2/ The main results of this study are provided by an in-house assay, making the comparison with other studies difficult. The authors say that “standard viscoelastic protocols are not sensitive to hypofibrinolysis and only sensitive to severe hyperfibrinolysis”.  However, such commercialized, semi-automated/automated, biological assay, easier to perform and allowing comparisons with other research teams, are used in similar studies with encouraging results (doi10.1186/s13613-019-0499-6; doi:10.1016/j.jcrc.2017.09.183; doi: 10.1016/j.bja.2020.12.010), please could you clarify or comment? 

Minor comments.

1/ I could not find in the “materials and methods” part the day the blood is sampled for analysis. I understand the samples are taken at day 1 after admission to ICU when reading the original paper but it may be precised in the materials and methods  (page 9, line 298).

2/ Page 9, line 316: the analysis are performed in duplicate: which value does the authors consider for statistics: mean value of the duplicates?

3/ In general, the fact that fibrinolytic capacity cannot be assessed by routine biological markers, notably by fibrin d-dimer, could be highlighted a little more.

4/References:

  • Page 12, line 8, reference 8: incomplete list of authors. Plus, this reference may not be relevant to illustrate that “The fibrin clot formation and lysis (clot-lysis) assay is a plasma-based assay which 48 can detect both hyper- and hypofibrinolysis », reference 10 could be enough.
  • Page 12, line 423, reference 14: national or international guidelines on DIC management would be more suitable than a review.
  • Page 12, line 439, reference 21: this is not a suitable reference to illustrate that “standard viscoelastic protocols are not sensitive 257 to hypofibrinolysis and only sensitive to severe hyperfibrinolysis” because this letter to the editor does not mention nor comment the fibrinolysis assessment with TEG

Reviewer 2 Report

Larsen JB et al assessed fibrinolysis in 34 septic shock patients using a plasma-based fibrin clot formation and lysis (clot-lysis) assay investigating the association with other markers of haemostasis, DIC, organ dysfunction and mortality.

They found 3 distinct clot-lysis:  1) severely decreased fibrin formation (flat clot-lysis curve), 2) normal fibrin formation and lysis and 3) pronounced lysis resistance. Patients with abnormal curves had lower platelet counts (p=0.05), more prolonged aPTT (p=0.04), higher lactate (p<0.01) and a tendency towards higher SOFA scores (p=0.09) than patients with normal clot-lysis curves. Fibrinogen and fibrin d-dimer were not associated with clot-lysis profiles (p≥0.37).

They concluded that septic shock patients showed distinct and abnormal clot-lysis profiles which were associated with markers of coagulation and organ dysfunction.

General comment: The paper is interesting, well written and well supported by the literature. However, as the Authors wrote in Discussion (lines 234-251), other similar studies explored the fibrin clot properties and clot-lysis assay and reported similar results.

Specific comments

  • I have some concerns about the presented data and the discussion. The relationship between the severity of sepsis and coagulation alterations supports the relationship between indexes of sepsis severity (lactate, SOFA score) and platelets count. In turn, platelets count was related to pathological flat and lysis resistant patterns. However, it doesn’t make sense to evaluate the link between Flat (n 10 patients), Normal ( n 13 patients) and Lysis Resistant (n 11 patients) pattern and baseline characteristics (see Results subchapter Study population, or age, comorbidities lines 107-109), or clinical outcome (Table 3, see mortality). You must have an appropriate cases number. I suggest restraining the results to laboratory data.
  • You wrote in Title and on lines 281-283 that your patients were “septic shock patients” (ref. 24). However, looking at Table 2 and Figure 3, I see most patients did not reach the level of lactate >= 2mmol/L. Apart from the need for noradrenalin, I see you did non consider the lactate level. Please comment.
  • On lines 252-263 the Authors make some interesting considerations about the potential application of different tests next years. Do you think your test will play a role in this scenario?

Reviewer 3 Report

Fibrin network formation and lysis in septic shock patients

The study by Julie Brogaard Larsen et al. aimed to study fibrin formation and fibrinolysis in patients with septic shock using a global (plasma-based fibrin clot formation and clot-lysis) assay. They also evaluated association between lysis parameters and prognosis. The topic is of high interest as there is need in tests able to properly measure fibrinolysis. The research is accurate and the manuscript well-written. Unfortunately, the sample size included is too small to draw any meaningful conclusion.

These are my major concerns:

1) Introduction. The rationale to use a global assay to evaluate the hemostasis in septic shock should be better explained. Thromboelastometry has used for example (see for example Boscolo A et al.  Whole-blood hypocoagulable profile correlates with a greater risk of death within 28 days in patients with severe sepsis. Korean J Anesthesiol. 2020).

A recent updated review dealing with tests to measure fibrinolysis should be added: Development and application of global assays of hyper- and hypofibrinolysis. Ilich A et al. Pract Thromb Haemost. 2019

2) REults: pg. 5 line 126 “Post-hoc tests showed…” What did the Authors mean with post-hoc tests?

3) Methods: centrifugation protocol for coagulation tests was not mentioned.

Thrombin generation method should be explained a little more; in particular the concentration of tissue factor used should be included.

Round 2

Reviewer 2 Report

I have no further comments

Reviewer 3 Report

The observations raised were addressed.